# The Effect of the PVA/Chitosan/Citric Acid Ratio on the Hydrophilicity of Electrospun Nanofiber Meshes

**DOI:** 10.3390/polym13203557

**Published:** 2021-10-15

**Authors:** Zsuzsanna Czibulya, Attila Csík, Ferenc Tóth, Petra Pál, István Csarnovics, Romána Zelkó, Csaba Hegedűs

**Affiliations:** 1Biomaterials and Prosthetic Dentistry, Faculty of Dentistry, University of Debrecen, 98. Nagyerdei Blvd, H-4032 Debrecen, Hungary; ferenc.toth@dental.unideb.hu (F.T.); hegedus.csaba.prof@dental.unideb.hu (C.H.); 2Laboratory of Materials Science, Institute for Nuclear Research (ATOMKI), 18/c Bem Square, H-4026 Debrecen, Hungary; csik.attila@atomki.mta.hu; 3Department of Experimental Physics, Faculty of Science and Technology, University of Debrecen, 18/a Bem Square, H-4002 Debrecen, Hungary; pal.petra@science.unideb.hu (P.P.); csarnovics.istvan@science.unideb.hu (I.C.); 4University Pharmacy Department of Pharmacy Administration, Semmelweis University, 7–9 Hőgyes Street, H-1092 Budapest, Hungary; zelko.romana@pharma.semmelweis-univ.hu

**Keywords:** electrospinning, polyvinyl alcohol, chitosan, citric acid, hydrophilicity

## Abstract

In this study, scaffolds were prepared via an electrospinning method for application in oral cavities. The hydrophilicity of the fiber mesh is of paramount importance, as it promotes cell spreading; however, the most commonly used polyvinyl alcohol (PVA) and other hydrophilic fiber meshes immediately disintegrate in aqueous media. In contrast, the excessive hydrophobicity of the scaffolds already inhibits cells adhesion on the surface. Therefore, the hydrophilicity of the fiber meshes needed to be optimized. Scaffolds with different polyvinyl alcohol (PVA)/chitosan/citric acid ratios were prepared. The addition of chitosan and the heat initiated cross-linkage of the polymers via citric acid enhanced the scaffolds’ hydrophobicity. The optimization of this property could be followed by contact angle measurements, and the increased number of cross-linkages were also supported by IR spectroscopy results. The fibers’ physical parameters were monitored via low-vacuum scanning electron microscopy (SEM) and atomic force microscopy (AFM). As biocompatibility is essential for dental applications, Alamar Blue assay was used to prove that meshes do not have any negative effects on dental pulp stem cells. Our results showed that the optimization of the fiber nets was successful, as they will not disintegrate in intraoral cavities during dental applications.

## 1. Introduction

Today, tissue regeneration has become the focus of much interest. [1] Several possibilities are available to promote the regeneration of damaged tissues. One method to prepare a scaffold aid material is the electrospinning of dilute solutions of carefully selected bio- and biocompatible synthetic polymers. Electrospun nanofibrous mats can be used as a scaffold [2,3,4,5,6,7,8,9,10] because of their large surface area, and because their structures and surface morphology can be controlled by optimizing parameters such as the voltage of the power supply, the feeding rate, or the distance between the needle and the collecting plate in the electrospinning process [6,11,12].

The selected materials for designing nanofibers for biomedical applications must fulfill strict requirements beyond biocompatibility and biodegradability (“which refers to the ability of a biomaterial being degraded by enzymes and other bio-based reactions when placed in the biological system” [13]); for example, they also must be nontoxic, moderately hydrophilic, and possess an appropriate mechanical strength, geometry, pore size, pore interconnectivity, distribution pattern and—in the case of implants—must be sterile or sterilizable [3,11,14].

One of the most commonly used polymers in electrospinning is polyvinyl alcohol (PVA), which is a nontoxic, biocompatible, water-soluble polymer, usable as a carrier of additives and active agents. PVA is able to enhance new developing tissues. Although one of its biggest advantages is its water solubility (at 80 °C temperature), which enables fast and homogeneous scaffold preparation, this also becomes its biggest disadvantage, as the fiber net is prone to dissolving during dental application because, without cross-linkage, this fiber mesh dissolves immediately [15,16]. Therefore, the structure of PVA fiber was modified by the addition of a more hydrophobic biopolymer, such as chitosan. There are advantages of using natural polymers, such as their ability to mimic the natural chemical environment, primarily by imparting biocompatibility, but there are also some drawbacks, as their weak mechanical properties result in fragile materials in comparison to synthetic polymers [17]. Therefore, it is appropriate to use a biopolymer/synthetic polymer mixture for the preparation of fiber-like meshes.

Chitosan is a biopolymer that has been widely investigated—for example, in bone regeneration—as its structure is very similar to that of glycosaminoglycan, which can be found in the ECM (extracellular matrix) [18]. The PVA/chitosan matrix’s applicability in a certain composition was tested by Pangon et al., using viability and proliferation measurements [19]. Shin et al. also found that a greater amount of bone was regenerated after application of a chitosan membrane compared to the control group [20]. Another application of chitosan combined with PVA films or fibers can be found in wound dressing [8,9,21,22,23], which is also supported by its broad antimicrobial spectrum. Therefore, the combination of these two polymers seems to be a good choice for the development of scaffolds for dental applications.

To stabilize the structure of prepared fiber mashes, and to decrease their solubility, a cross-linking agent can be added to the system, and the cross-linkage process can be initiated by heat or plasma treatment. For chemical cross-linking of polymers, glutaraldehyde (GA) is usually used. GA can be used for cross-linking PVA [24] or chitosan molecules [25], or both [26]. Although glutaraldehyde is a widely used agent, its respiratory and skin irritation were proven from extended or repeated exposure at low concentrations. Even trace amounts of GA in the scaffold can have these and other undesirable effects—for example, cytotoxicity [24,27,28]. A more adequate choice can be citric acid—a green cross-linking agent containing three carboxyl groups, whose antimicrobial and antioxidant properties are well known. These can take part in cross-linking reactions of different magnitudes [29]. Citric acid can be used to cross-link different polymers, including PVA [15,16,30] or chitosan (as this ionic cross-linker can increase the water-resisting capacity of chitosan films [31]), or even both of these polymers [19]. The cross-linkage can be initiated by heat treatment [32].

The thus-prepared and electrospun fiber net can support bone regeneration, and can be used as a carrier and deliverer of dental materials—for example, biologically active ingredients, factors, or drugs if its composition is adequate—provided its stucture is rigid enough and its hydrophilicity and related water solubility are optimal. In such a case, it maintains its structure and pore size when contacting water, until the tissue regeneration reaches the point when no support is needed. The aim of this study was to prove that, in addition to the adequate cross-linkage [33], the PVA/chitosan/citric acid ratio can have an effect on the applicability of the fiber net during dental applications in aquaous media, as it will not dissolve in water during the application.

## 2. Materials and Methods

### 2.1. Materials

For the preparation of the polymer mixture, polyvinyl alcohol (PVA 18-88. Merck, Darmstadt, Germany, degree of hydrolyses according to USP 85–89%) of active pharmaceutical ingredients quality was used; its molar mass was determined via gel permeation and size-exclusion chromatography (Agilent Technologies 1260 Infinity II chromatograph, Phenomenex PolySep GFC-P Linear, California, USA 300 mm × 7.8 mm, Separation range 1 KDa–10 MDa) colonna containing water with 5 mM perchloric acid and 30 *v/v*% acetonitrile as eluents at 40 °C. The flow rate of eluent was 0.7 mL/min, and a differential refractive index (ri) detector was used. The molar mass was found to be ~65,000, with 3.168 as the degree of polydispersity. Before fiber preparation, the PVA was left to swell in distilled water under mixing, and was kept at 80 °C until dissolving. Chitosans’ (28,191, from crab shells, Sigma-Aldrich, St. Louis, MO, USA; the batch used by us was similar to that of Lai et al., 2017 [34]) molecular weight was determined via size exclusion chromatography against polyethylene oxide molecular weight standards on a Phenomenex Polysep-P-Linear (300 mm × 7.8 mm) column using 0.1% aqueous trifluoroacecic acid as an eluent. The system consisted of a Merck-Hitachi LaChrom D-7000 liquid chromatograph equipped with variable wavelength UV and differential refractive index detectors. The eluent flow rate was set to 0.7 mL/min, and the column compartment was thermostated to 40 °C. Chitosan was found to have a peak molecular weight of 750–1000 kDa, and broad polydispersity. The degree of deacetylation was calculated from the **^1^**H-NMR spectrum of 10 mg of chitosan dissolved in 1 mL of D**_2_**O containing 0.08 M DCl, using the formula of Hirai et al. (Polymer Bulletin 26, 87–94, 1991). The chitosan sample was found to have a degree of deacetylation of 82%. Chitosan was dissolved in 2 *v/v*% acetic acid solution at 80 °C under mixing.

In the case of the initial polymer mixtures, the amount of the PVA precursor was fixed, and the added chitosan content was increased. The *m/m*% ratio of PVA in the solution varied from pure PVA to 2 times that of chitosan’s *m/m*%. The amount of citric acid was chosen similarly to that of Pangon et al.; based on their work [19], we tried to find the optimal amount of cross-linker considering that the chitosan solution was prepared in 2 *v/v*% acetic acid, which also contains COOH groups. The sample composition can be seen in Table 1 in Section 2.2.

The sample preparation for electron microscopy analysis was performed as described previously [35]. Briefly, the cell–containing samples were fixed with 2 *v/v*% glutaraldehyde for 2 h and at 1 *m/m*% OsO_4_, and then dehydrated using graded ethanol solutions, and their critical points were dried using CO_2_ before the examination.

### 2.2. Methods

#### 2.2.1. Determination of the Viscosity of Polymer Solution Mixtures

The viscosity of the polymer mixtures was determined from flow curves using an MCR 102 Rheometer (Anton Paar GmbH, Graz, Austria), with a measuring system consisting of a 50 mm parallel plate at 25 °C, by increasing and then decreasing the shear rate ramp via a Bingham equation, as the mixture was expected to show plastic properties. In Equation (1), *τ* represents the shear stress, *τ_B_* is the Bingham yield stress, and *η_pl_* is the plastic viscosity [36].
(1)$$\tau =\tau_{B}+\eta_{pl}\left(d\gamma |dt\right)$$

#### 2.2.2. Electrospinning

The fibers (native samples: F_1_ to J_1_) with increasing chitosan and citric acid content were prepared using a Nanospinner NS1 device (Inovenso Ltd., Istambul, Turkey). Solutions were used to fill a 5 mL volume syringe, to which the metal capillary was connected by a silicon tube. The electrospun nanofibers were deposited on the 15 cm × 15 cm grounded collector electrode covered with aluminum foil, placed 13 cm away from the top of the capillary at a 25 kV voltage. The flow rate was fixed at 0.65 mL/h (Alaris GH infusion syringe pump). The deposition time was 30 min.

For the cell viability measurements, round glass cover slides of 1.3 cm diameter were stuck on the aluminum foil, and electrospinning was performed on them using the same settings as before. The samples were treated with UV light for sterilization.

The composition of the samples and the indication of heat treating can be seen in Table 1.

To initiate cross-linkage, samples were heat-treated by using heating plate (heat-treated samples: F_2_ to J_2_) at a 120 °C temperature for 4 h. Regarding our preliminary results, due to the addition of PVA to the system, the fibers prepared from the mixture did not melt at up to 120 °C, which was the lowest temperature to be able to induce cross-linkage. Below this temperature, the cross-linkage was unsuccessful, as the fibers simply dissolved in one drop of water.

An example of the thus-prepared fiber net can be seen in Figure 1.

#### 2.2.3. Study of Composition and Monitoring of Cross-Linking Reactions via Infrared Spectroscopy

Infrared spectra were measured using the Smart iTR™ Attenuated Total Reflectance (ATR) Sampling Accessory of a Nicolet 6700 FT-IR Spectrometer (Thermo Fisher Scientific, Waltham, MA, USA). Samples were scanned 16 times at 0.482 cm^−1^ resolution, using Diamond ATR Crystals specifications. The measured spectra were deconvoluted using the OriginPro program.

#### 2.2.4. Morphological Study via Low-Vacuum Scanning Electron Microscopy

The morphological study of fiber samples was carried out via low-vacuum scanning electron microscopy (LV-SEM, JEOL IT500HR, Tokyo, Japan). To avoid the charge accumulation of non-conductive samples, the measurements were implemented at a low accelerating voltage (3 kV) and in high-purity nitrogen at 30 Pa.

#### 2.2.5. Morphological Study via Atomic Force Microscopy

Topology and roughness were measured via atomic force microscopy (AFM; AFM-Raman HR Nano, HORIBA Jobin Yvon S.A.S., Longjumeau, France) using cantilevers—HQ: NSC14/Al Bs at 160 kHz, 5.0 N/m. All samples were measured 3 times at 3 magnifications (20 × 20, 10 × 10, and 5 × 5 microns), the resolution was 256 × 256, and the velocity of scanning was 0.3 Hz.

#### 2.2.6. Morphological Study via Contact Angle Measurements

The wettability of the fibers by water was characterized using contact angle measurements performed with a Krüss Drop Shape analyzer (Krüss GmbH, Hamburg, Germany), applying the Young–Laplace model. The measurement points determined up until the 5th second after placing the water drop onto the surface were used for evaluation in order to avoid inaccuracies arising from the evaporation. The size of the drops was 0.004 mL, and the number of measurement repetitions was 5.

A measurement example can be seen in Figure 2.

#### 2.2.7. Determination of the Swelling Ratio

The swelling ratio was calculated from measurements of the mass of the samples before (*Wd*: weight of dry samples) and after being wetted (*Ws*: weight of swollen samples) for a week with distilled water. The wet fiber net samples’ mass was measured after removing the excess water with filter paper at room temperature. Swelling was monitored for 8 days. The swelling ratio was calculated using Equation (2) [30,37], and was given for the 7th day, as the values reached their maximum:(2)$$degree\;of\;swelling\left(\%\right)=\left(\frac{\left(Ws-Wd\right)}{Wd}\right)*100$$

#### 2.2.8. Cell Culture and Viability Assay

Dental pulp stem cells (DPSCs) were cultured in DMEM F12 medium (Thermo Fisher Scientific, Waltham, MO, USA) supplemented with 10 *v/v*% fetal bovine serum, 1 *v/v*% GlutaMAX, and 1 *m/m*% antibiotic–antimycotic (all from Thermo Fisher Scientific), and incubated in a humidified incubator at 37 °C in 5 *v/v*% CO_2_. A total of 10^5^ cells were seeded onto the surface of different chitosan-containing, fiber-covered glass coverslips placed in 24-well plates and incubated under the same conditions. Cells grown on the surface of PVA-fiber-covered (without chitosan, see Table 1 for concentrations) glass coverslips served as positive controls for the experiment. Three parallels were set for each group. The viability of the cells was determined using the Alamar Blue assay (Thermo Fisher Scientific). After 1 and 4 days, the cell culture medium of each well was replaced with 10 *v/v*% of Alamar Blue reagent. Following 1 h of incubation at 37 °C in the dark, the fluorescence of the samples was measured using a microplate reader (HIDEX Sense Turku, Finland) at 544 nm excitation/595 nm emission.

#### 2.2.9. Vitality Staining

DPSCs cells (10^5^/well) were seeded on UV-light-treated (30 min) fiber-covered glass coverslips, and then incubated for 4 days. PVA fiber-covered coverslips were used as controls. After the incubation period, the cells were co-stained with fluorescein diacetate (FDA) and propidium iodide (PI) (both from Sigma-Aldrich) for 5 min at room temperature. Pictures were taken using a Zeiss AxioVert A1 inverted fluorescence microscope (Carl Zeiss Microscopy GmbH, Jena, Germany).

#### 2.2.10. Statistics

All statistical calculations were performed using Microsoft Office Excel. The significance of the determined parameters was studied via T-probe. Fiber diameter parameters for roughness were calculated from SEM micrographs using the ImageJ program. Fiber diameter was determined from SEM micrographs using ImageJ software, measuring 100 pieces of fibers; the average diameter was calculated using this program, and from AFM pictures derived from its software, together with the roughness Ra value. The distribution was calculated using Excel, and was plotted using the OriginPro program.

## 3. Results

### 3.1. Investigation of Initial Polymer Mixtures

The viscosity of the initial polymer mixtures can be found in Table 2.

According to preliminary experiments, the closer the viscosity of the mixture approaches to that of PVA, the easier the electrospinning can be performed. Too high viscosity and the higher chitosan content than the optimal value can cause inhomogeneities occurring in the fiber net.

### 3.2. Investigation of the Structure of the Fiber Network via Scanning Electron Microscopy

According to our preliminary results, high chitosan content can be one of the causes responsible for spindles and beads occurring in the fiber net, as the viscosity of the polymer solution becomes too high for electrospinning; this can be seen in the SEM images (Figure 3a,b) in Sample I_1,2_ and J_1,2_ containing 1.9–2.1% chitosan. Heat-treated samples showed similar morphology to native samples. Microscope images of the native and heat-treated fiber net samples show that the density of the fibers also decreased with increasing chitosan content.

Regarding the SEM images, it can be seen that the average fiber diameter of native and heat-treated samples (F_1,2_, G_1,2_, H_1,2_, I_1,2_, J_1,2_) decreased with increasing chitosan and citric acid content from F to I. The full-width values (Table 3) of distribution curves followed a minimum curve in the case of native samples, and decreased with increasing chitosan and citric acid content in the case of heat-treated samples.

The heat treatment caused the following changes: In the case of the pure PVA sample (F_1_), the maximal fiber diameter distribution shifted to the left (F_2_), and the distribution narrowed—probably due to conformational transitions or the release of adsorbed water due to heat [15]. In the case of the chitosan-containing samples, the effect of the heat treatment could not be clearly observed by evaluating the SEM images since, due to the low vacuum, the fibers became charged, and the stuck fibers could not be differentiated from the others; therefore, AFM measurements were also performed (Figure 4).

### 3.3. Investigation of the Structure of the Fiber Network via Atomic Force Microscopy

Although the measurement method was different, and the calculated values of the average fiber diameters from the SEM images should be more precise than those calculated from the AFM images, the trend was found to be similar. (Table 4) In the latter case, even the effect of the heat treatment could be more clearly observed, as the charging of the fibers did not disturb the evaluation. The fiber diameters decreased significantly with increases in the chitosan and citric acid content, and increased in response to the heat treatment, as some of the fibers became stuck (comparing native samples (F_1_, G_1_, H_1_, I_1_, J_1_) to the related heat-treated samples (F_2_, G_2_, H_2_, I_2_, J_2_)). Szewczyk et al., found that some of the surface roughness parameters—for example, *Ra*, the value of which could also be calculated from SEM images using ImageJ software—correlated well with the contact angle parameters [38], but we found that the thus-calculated data fluctuated. To avoid this failure arising from manually attributing area values to the ImageJ software, roughness parameters were determined from AFM images with the same software.

The roughness (*Ra*) value of the samples measured by AFM showed a decrease with the increase in the chitosan and citric acid contents of the samples, meaning a decrease in the density of the fiber net. Even in the case of heat-treated chitosan-containing samples (Samples G_2_ to J_2_), this parameter showed decreasing values due to heat treatment, likely because the fibers lost their moisture content.

### 3.4. Interpretation of Contact Angle Measurements

The optimization of hydrophilicity via appropriate mixture composition was followed by contact angle measurements using the Krüss Advance Drop Shape Analyzer.

The measured contact angle values of the serial measurements can be seen in Table 5. The measurement of the native samples’ contact angle with water was problematic, because the samples dissolve in water quite fast. Therefore, their values are not indicated here.

The contact angles of the heat-treated samples increased with increasing chitosan and, therefore, increasing citric acid content (when comparing the samples by increasing chitosan/citric acid content, only in the case of comparing H_2_ to I_2_ was no significant change found); this was accompanied by a decrease in the Ra values calculated from AFM images (As G_2_ contained the lowest chitosan/citric acid content, it could not be differentiated well from the control (F_2_ sample). To confirm the development of cross-linkage, infrared spectra were measured.

### 3.5. Infrared Spectroscopy Measurements

The increase in hydrophobicity related to cross-linking was also proven via infrared spectroscopy measurements (Figure 5).

Fourier-transform infrared spectroscopy [31,39] was conducted to identify the cross-linking reaction between chitosan and PVA initiated by the addition of citric acid and heat treatment. Assuming that this reaction also begins without heating, the native fiber’s infrared spectra were investigated and then compared to heat-treated fibers. As the investigated system was very complex, and the presence of water could not be avoided, only increases in the two peaks related to cross-linkage by citric acid were investigated: one related to the amide II group around 1550 cm^−1^ (associated with the N-H bending vibration bond of amide II aromatic nitro compounds), and one related to the developed ester bond around 1720 cm^−1^ (Table 6). To make them comparable, the spectra were normalized to the 843 cm^−1^ peak related to the aromatic C-H in-plane and out-of-plane bending and the vinyl C-H out-of-plane bonds, and were deconvoluted into peaks using OriginPro software.

Regarding the data, it can be said that the peak related to the amide II bond (1552 to 1573 cm^−1^) also increased with increasing chitosan content in the case of both native and heat-treated samples. It can also be seen that, without heat treatment, the cross-linking reaction could not completely go through, as in the case of native samples (F_1_, G_1_, H_1_, I_1_, J_1_), both investigated peaks—the one related to the amide II bond, and the peak around the ester carbonyl bond—were present, but they became fully expressed after heat treatment (F_2_, G_2_, H_2_, I_2_, J_2_) due to more complete cross-linkage.

### 3.6. Swelling Studies

To prove that the heat-treated fibers do not dissolve in water after heat treatment as the native ones did (as can be seen in Figure 6), samples were immersed in water for 8 days.

The swelling was monitored for 8 days. The swelling ratio was calculated, and was given for the 7th day, as the values reached their maxima at this point; results are shown in Figure 7.

The samples did not dissolve during the treatment and, after drying, the structure could be investigated via scanning electron microscopy. In the case of the swelling values, significance could not be asserted except for the high-chitosan-content samples (I_2_–J_2_), as the standard deviation was extremely high because of the irregularity of the bulk fiber samples from which the samples were cut. This characteristic of the fibers was balanced out by the formation of spindles and beads, as these samples (I_2_ and J_2_) showed more determined values. A significant decrease in swelling properties was found with increasing chitosan concentration. Aside from the high standard deviation of the swelling results, regarding the contact angle values, a large-scale change was found from 28.47° to 42.58°, from which it can be said that H_2_ samples have the optimal physical properties for application in the oral cavity.

As the native PVA-containing samples’ structure decomposed during contact angle measurements, preliminary experiments of structure changes for cell viability were performed in order to prove that the structure of heat-treated samples can resist water for the adequate length of time. Therefore, the samples F_2_* (heat-treated PVA, * means: added citric acid to promote cross-linkage, as pure PVA dissolves immediately in water) and H_2_ (the one containing 1.3 *m/m*% chitosan with fiber-like structure) were immersed in distilled water for 1–5 days, and SEM images were taken in order to see whether the samples dissolved in water or not, and to study their structural changes. The SEM images showing this can be seen in Figure 8.

It can be seen that the morphology of F_2_* changed drastically even after 1 day of immersion in water, while the H_2_ sample had only slightly changed after 1 day of water treatment (Pangon et al., 2016 found a similar change in their fiber structure [19]). The fiber-to-film transition could be observed due to the plasticizing effect of the absorbed water on the F_2_* sample after 1 day, and on the H_2_ samples after 5 days of soaking.

### 3.7. Biocompatibility and Viability Results

The viability of DPSC cells on the surface of UV-treated fiber-covered glass coverslips was determined via Alamar Blue assay after 1 and 4 days to examine the biocompatibility of the samples. We aimed to determine whether the optimized polymer mixture’s components have any effect on the viability or proliferation of the cells grown on the heat-treated fiber net (samples F_2_ to J_2_). The viability of the cells grown on the G_2_, H_2_, and I_2_ chitosan-containing fiber-covered coverslips changed slightly but significantly (Figure 9) after 1 day of incubation, while on sample J_2_ it was unchanged compared to the control (F_2_, PVA). After 4 days, however, none of the chitosan-containing samples caused significant alterations in the cell viability.

It was shown that, in contrast to native samples, the heat-treated fibers were not damaged due to water and chemical treatments, and the cells entered the fiber net (Figure 10).

Thus, the physically/chemically optimal fiber sample containing the most chitosan without forming spindles and beads (Sample H_2_) can be used as a carrier during dental applications.

## 4. Discussion

Using a nanofiber net prepared via electrospinning as a holder can be one solution to induce tissue regeneration. Therefore, polymer solutions with optimized compositions had to be prepared. Nanofibers prepared from polyvinyl alcohol mixed with chitosan can be used as a scaffold, as they are biocompatible with human cells—in our case, with bone cells [19]. According to their consumption, their physical and chemical parameters must also be optimized. The composition of the initial polymer mixture is one of the most important parameters in fiber net preparation, as it affects the water solubility of the fiber net [41]. It is well known that pure PVA has excellent solubility [42] in hot water, which is an advantage in solutions prepared in aqueous media; however, the fiber mesh without any structural stabilization can dissolve in aqueous media immediately. For dental applications, it is essential that the prepared scaffold can maintain its structure for an adequate time. This can be promoted by adding chitosan and adequate amounts of citric acid to the system, and then initiating the cross-linkage of the two polymers via heating. [15]. Pangon at al. showed that multicarboxylic acids present at high percentages (10% or above) can be used for cross-linking PVA and chitosan; they also found more expression of hydrophobicity in the thus-modified fibers. [19] Heat treatment was also shown to increase the water insolubility of the PVA/chitosan fibers by Riwu et al., but the combination of the effects of these modifiers, the heat treatments, and the addition of green cross-linking agents (below 5%) have not yet been investigated [32].

The difficulty is that the increase in chitosan content can significantly increase the viscosity of the polymer mixture (for example, from 3 mPas (cP), as Pervez et al. found [41], up to 1300 mPas, as was determined in our work). The initial viscosity of the polymer mixture solution is one of the parameters that is responsible for the adequate structure of the fiber net and the optimal fiber formation. In our case, the applicable viscosity was found to be in the range of 500–1000 mPas. The formed fiber structure can be studied via two methods: low-vacuum scanning electron microscopy, and atomic force microscopy. We used both methods, as ImageJ software cannot correctly distinguish the stuck fibers from broadened ones on SEM images, so the measured values had to be confirmed using AFM images. In our work, the maxima of the fiber diameter distribution curves were found to be around 200 nm according to both methods. The results of both methods showed that the diameter of the fibers decreased with increasing chitosan and citric acid content, and increased due to heat treatment. Although, according to Yao et al., fibers with a beaded structure can also have some medical potential in drug delivery—which merits further investigation [43]—our purpose was to prepare structured fiber samples; therefore, the two samples with the highest chitosan content—I_1,2_ and J_1,2_—were not used for further investigation.

As the cell adhesion and growth can be promoted by moderately hydrophilic domains [3], our aim was to develop an adequate hydrophobic fiber net that retains its structure during the application.

To investigate the hydrophobicity of our samples, several parameters can be used. Szewczyk et al. found [38] that some roughness parameters (which they calculated from SEM images using ImageJ software) can correlate well with the contact angles, but it is well known that the calculated roughness depends on the area, which we had to set manually. To avoid this error, roughness parameters were measured via atomic force microscopy, and the measured parameters are shown in our study. Regarding heat-treated samples, a linear correlation was found between decreasing *Ra* and increasing contact angle.

The increase in hydrophobicity can be proven regarding the increase in the contact angles of the heat-treated samples, which in our study changed around 25–53° due to different amounts of cross-linker addition; Perves et al. found a larger range of this change: 14.68–64.74° [41]. As was proved by Sathypathy et al., in the case of other ratios of chitosan to PVA, in special cases, the contact angle can even decrease (as in the case of our I_2_ sample) because of the availability of the OH group in the saccharide moieties in chitosan [44]; however, this does not mean that the sample’s solubility in water would increase. Cui et al., in 2017, showed that two parallel processes can take place: the increase in chitosan content decreased the contact angles, but the cross-linking with glutaraldehyde increased the hydrophobicity of the net [26]. This process could also be induced by additional heat treatment [15,32] or plasma treatment [45]. This is also true in the case of the green cross-linker—citric acid [46]—which was chosen in the present work [47].

Cross-linking process can also be monitored via infrared spectroscopy, as there is an indirect correlation of crosslinking with hydrophobicity and higher contact angles. In the case of fibers under certain roughness conditions, air bubbles can be trapped, and the contact angles increase, so only apparent contact angles can be given. This property can be described by the Cassie–Baxter state [48]. If one cannot avoid moisture in the sample preparation and measurements, one cannot quantitatively compare all of the peaks, such as that at 3500–3200 cm^−1^ [49]. This is a peak linked to the stretching of O-H from the intermolecular and intramolecular hydrogen bonds, and can be broadened by the presence of the NH-group of chitosan, due to which the cross-linking process must decrease. Chitosan can also be cross-linked via citric acid. The absorption bands at 1561 cm^−1^, 1658 cm^−1^, and 1320 cm^−1^ correspond to amide II, amide I, and amide III bands, respectively, and are characteristic of chitosan. The bands at 1573 cm^−1^ and 1530 cm^−1^ are related to cross-linked chitosan, while that at 1715 cm^−1^ is related to to citric acid–chitosan bonding [31,50]. Nascimento showed that the cross-linking of the PVA membrane with citric acid caused an increase in the peak around 1720 cm^−1^, characteristic of the esterification reaction between the polymer hydroxyl and the citric acid carbonyl [46]. This characteristic peak for the cross-linking of PVA and chitosan was also found by Pervez et al., as they showed that the peaks between 1700 and 1725 cm^−1^ characterize the dimers of carboxylic acid (also found by Das et al., 2018 [50]). In our work, the change in this peak proved the development of cross-linkages, as well as the hydrolysis of the unhydrolyzed acetyl groups of PVA [40].

The swelling behavior of the nanofibrous net also plays an important role in cellular adhesion and proliferation, as it influences the fluid intake capacity of the scaffold. Satpathy et al. found a swelling ratio (%) of 130.18% for PVA, and 142.42% for PVA/chitosan [44]. In our study, with a higher chitosan amount, we found ~3–5 times higher swell ratio values.

The examination of the cell viability showed that none of the chitosan-containing PVA fibers affected the viability of the cells, compared to the control, and the cells were able to proliferate onto all of these samples. These observations suggest that the fibers are not cytotoxic and, thus, are possibly biocompatible. However, the changes in the physical parameters of the different chitosan-containing samples affected the morphology of the cells, most likely through the changes in the attachment possibilities on the fiber surfaces.

Regarding the physicochemical studies, the heat-treated sample (H_2_) contained the most applicable (1.3 *m/m*%) chitosan content, as it had the required fiber-like structure (without any beads or spindles, unlike samples I_2_ or J_2_), and its contact angle related to its hydrophilicity was also found to be optimal (not so low as that of G_2_) for cell growth. As this scaffold did not significantly affect the viability of the DPSC cells, it is possibly biocompatible and, thus, suitable for dental application.

## 5. Conclusions

In thus work, we proved that in addition to the degree of cross-linkage, the PVA/chitosan/citric acid ratio also has an effect on the hydrophilicity and the cell viability—and, thus, on the biocompatibility—of the fiber net. We successfully optimized the chemical composition and the cross-linkage of the fiber net, achieving appropriate hydrophilicity (H_2_ sample with contact angle: 42.5 ± 4.29°), as according to our results the original structure of the heat-treated scaffolds likely will not change significantly in aqueous media during the working and curing time. As the mesh (H_2_ containing 6.5 g/100 g PVA, 1.3 g/100 g chitosan and 2.9 g/100 g citric acid) maintained all of the necessary physical and chemical properties (for example: geometry, pore size, pore interconnectivity) of nanofibers, and proved to be biocompatible with DPSC cells, it can potentially be used for dental applications.

## Figures and Tables

**Figure 1 polymers-13-03557-f001:**
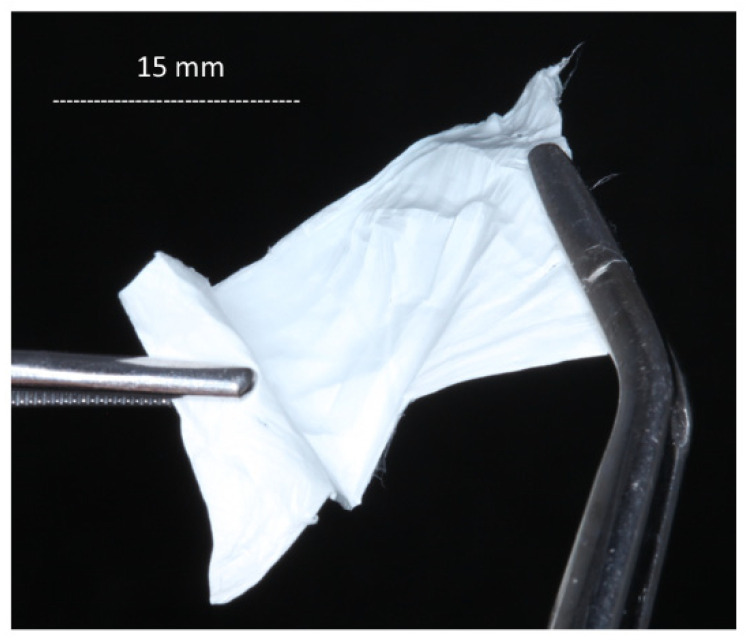
A fiber net prepared via the electrospinning method.

**Figure 2 polymers-13-03557-f002:**
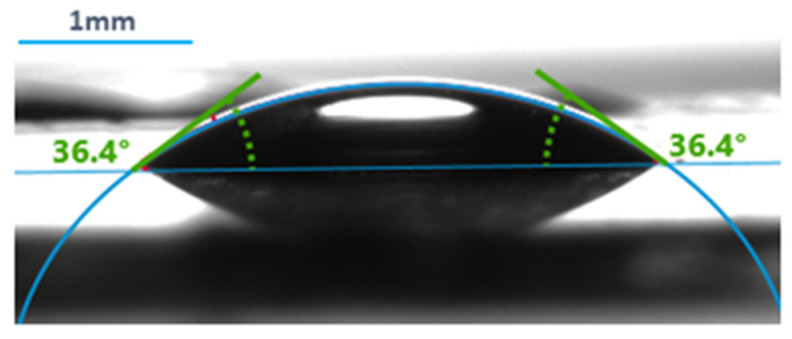
An example of contact angle measurement using the Krüss Advance Drop Shape Analyzer.

**Figure 3 polymers-13-03557-f003:**
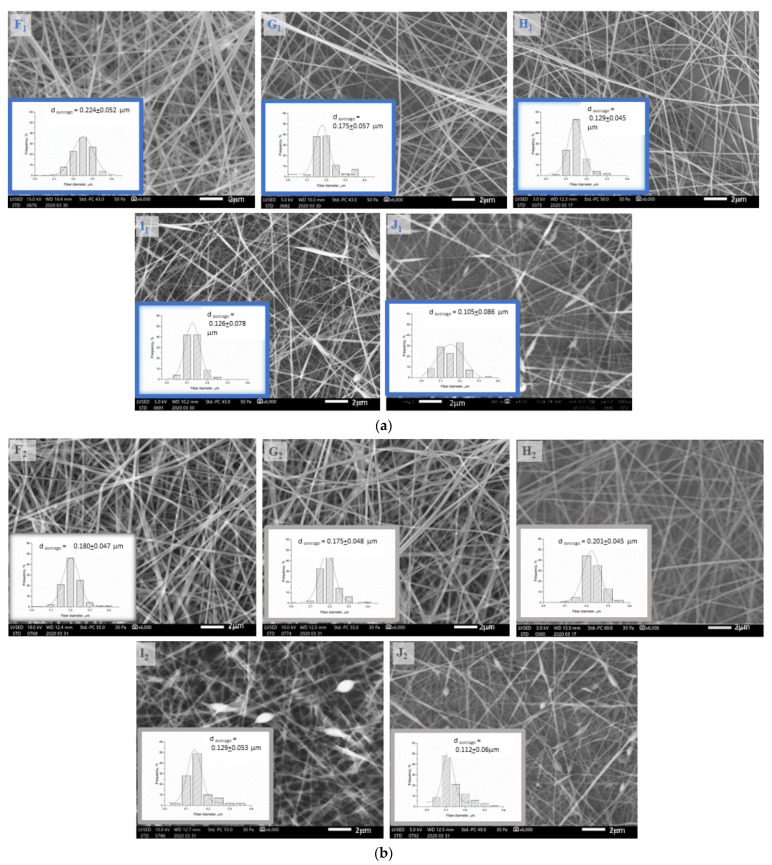
(**a**) The scanning electron microscopy images and distribution curves of native (F_1_, G_1_, H_1_, I_1_, J_1_) fiber samples taken at 6000× resolution. (**b**). The scanning electron microscopy images and distribution curves of heat-treated (F_2_, G_2_, H_2_, I_2_, J_2_) fiber samples taken at 6000× resolution.

**Figure 4 polymers-13-03557-f004:**
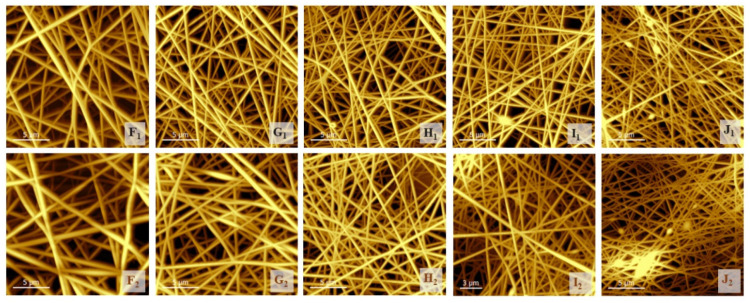
The atomic force images of native (F_1_, G_1_, H_1_, I_1_, J_1_) and heat-treated (F_2_, G_2_, H_2_, I_2_, J_2_) samples.

**Figure 5 polymers-13-03557-f005:**
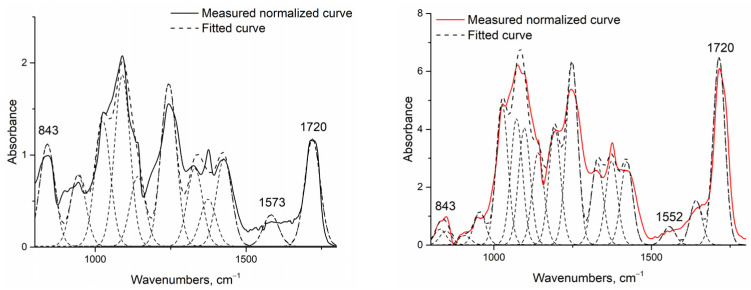
Example of normalized and deconvoluted IR spectra of H_1_ (**left**) and H_2_ (**right**) samples.

**Figure 6 polymers-13-03557-f006:**
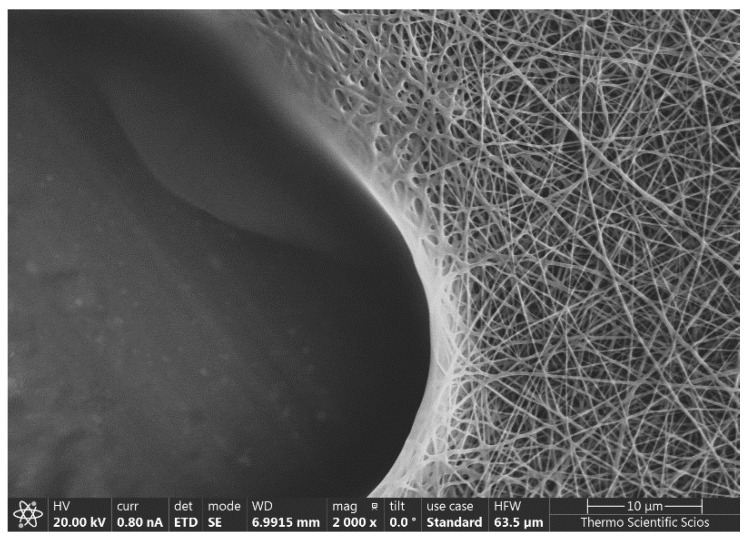
An example of an SEM image of a native sample (G_1_) after placing one drop water on the surface and allowing it to dry in air, at room temperature.

**Figure 7 polymers-13-03557-f007:**
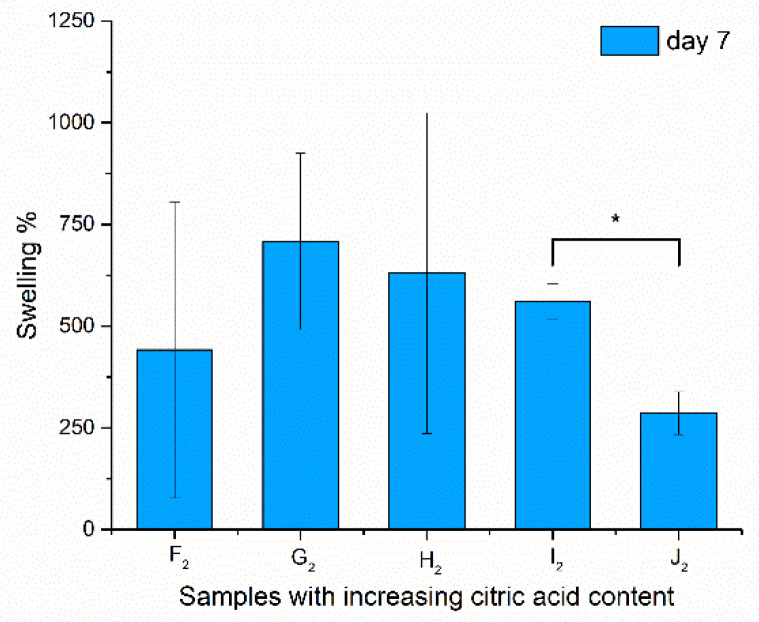
Swelling properties of the samples after being immersed in water for 7 days (due to high standard deviation, no significance was found at *p* > 0.1, but I_2_–J_2_ are indicated with *).

**Figure 8 polymers-13-03557-f008:**
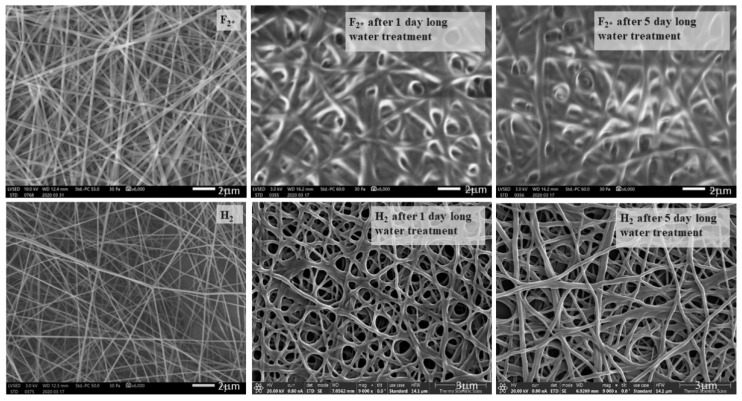
Scanning electron microscopy images of air-dried F_2_* and H_2_ samples immersed in water for 1–5 days, taken at 6000× and 9000× resolution, with 2 μm scale, and 3 μm in the case of H_2_ samples after water treatment.

**Figure 9 polymers-13-03557-f009:**
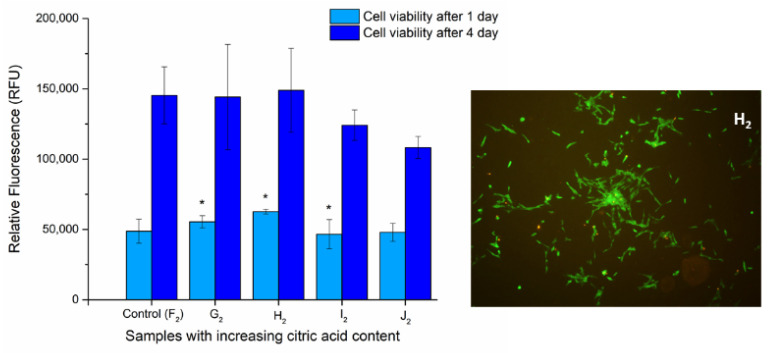
Cell viability assay of DPSC cells: Cells were cultured on the surface of fiber-covered glass coverslips for 1 and 4 days. After incubation, cell viability (**left**) was measured by Alamar Blue assay. Values are expressed as sample means; * means samples significant to one another, while error bars represent the standard deviation (SD) of three parallel measurements (*p* < 0.05). Vitality staining of MG63 cells (**right**): The cells were seeded onto the fiber-covered coverslips and cultured for 4 days. After the incubation period, cells on the different surfaces were co-stained with fluorescein diacetate and propidium iodide.

**Figure 10 polymers-13-03557-f010:**
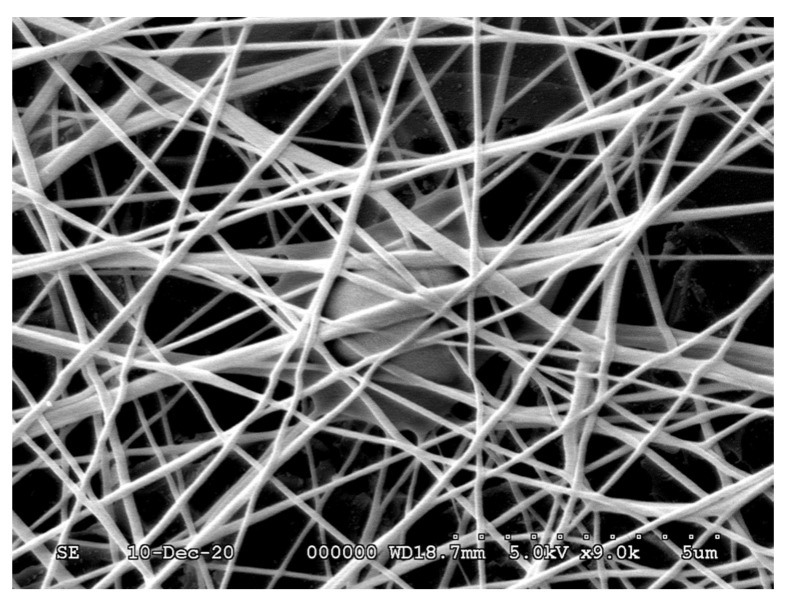
Scanning electron microscopy image of an H_2_ sample containing DPSC cells, at 9000× resolution.

**Table 1 polymers-13-03557-t001:** Composition of the initial polymer mixtures.

Sample	PVA, g/100 g	Chitosan, g/100 g	Citric Acid, g/100 g	Sample Modification
F_1_ (PVA)	10.0	0.0	0.0	native
F_1_ * (PVA)	10.0	0.0	6.1	native
F_2_ (PVA)	10.0	0.0	0.0	heat-treated
F_2_ ** (PVA)	10.0	0.0	6.1	heat-treated
G_1_	7.8	0.8	1.8	native
G_2_	7.8	0.8	1.8	heat-treated
H_1_	6.5	1.3	2.9	native
H_2_	6.5	1.3	2.9	heat-treated
I_1_	4.8	1.9	4.2	native
I_2_	4.8	1.9	4.2	heat-treated
J_1_	4.2	2.1	4.7	native
J_2_	4.2	2.1	4.7	heat-treated

*,** In the sample names refers to the presence of citric acid.

**Table 2 polymers-13-03557-t002:** Calculated viscosity (*η_pl_*, mPas) values from flow curves. All *η_pl_* values increased significantly compared to the PVA (*p* < 0.05), and all data increased significantly compared to one another (*p* < 0.1), * after the samples name except for I and J.

Sample Name	*η*_*pl*_, mPas	St. dev
F_1,2_ (PVA)	532	28
F_1,2_ *	490	25
G_1,2_	905	51
H_1,2_	1050	73
I_1,2_ *	1243	88
J_1,2_ *	1317	106
Chitosan	3561	209

**Table 3 polymers-13-03557-t003:** The full-width at half-maximum values of fiber diameter distribution (a supplement of Figure 3).

Samples	Full-Width at Half-Maximum Values of Distribution	St. dev.
F_1_, (PVA)	0.1280	0.0084
F_2_, (PVA)	0.0989	0.0015
G_1_	0.0829	0.0144
G_2_	0.0937	0.0074
H_1_	0.0815	0.0035
H_2_	0.0919	0.0135
I_1_	0.0844	0.0091
I_2_	0.0753	0.0067
J_1_	0.1737	0.0586
J_2_	0.0673	0.0119

**Table 4 polymers-13-03557-t004:** Calculated average fiber diameter values (*d_ave_*., μm) and surface roughness parameters (*Ra_ave_*_,_ μm) from AFM images of native and heat-treated samples.

Sample	*d*_*ave.*_ μm	St. dev.	*Ra*_*ave.*_ μm	St. dev.	*Ra*_not significant_*p* > 0.1
F_1_	0.230	0.005	0.353	0.033	
F_2_	0.252	0.005	0.358	0.090	vs. F_1_
G_1_	0.175	0.020	0.283	0.043	
G_2_	0.236	0.002	0.254	0.035	vs. G_1_
H_1_	0.150	0.040	0.230	0.018	
H_2_	0.192	0.007	0.150	0.013	
I_1_	0.116	0.030	0.221	0.040	vs. H_1_
I_2_	0.153	0.003	0.142	0.023	vs. H_2_
J_1_	0.101	0.050	0.179	0.016	
J_2_	0.124	0.003	0.121	0.046	vs. I_2_

**Table 5 polymers-13-03557-t005:** Contact angles of serial samples.

Samples	Contact Angle, °	St. dev.	Not Significant *p* > 0.1
F_2_, (PVA)	25.4	2.6	
G_2_	28.5	1.9	vs. F_2_
H_2_	42.6	4.39	
I_2_	41.2	4.7	vs. H_2_
J_2_	51.8	4.3	

**Table 6 polymers-13-03557-t006:** The fitted IR peaks corresponding to amide II and ester bonds.

Samples/Peaks	1552 to 1573 cm^−1^ Amide II Bond	St. dev.	1720 to 1735 cm^−1^ Peak of Ester Bond	St. dev.
F_1_ (PVA)	0.0000	0.0000	0.0000	0.0000
F_2_ (PVA)	0.0000	0.0000	0.0000 or co-peak *	0.0000
G_1_	0.1756	0.0487	0.8028	0.1834
G_2_	0.1470	0.0258	1.4082	0.2887
H_1_	0.1953	0.0275	1.1657	0.1802
H_2_	0.3555	0.1016	6.2583	1.8627
I_1_	0.6716	0.1211	2.9992	0.3213
I_2_	1.5974	0.2282	6.2351	0.2024
J_1_	0.7784	0.2724	2.2367	0.3972
J_2_	2.4740	0.4948	17.0395	2.6624

* In the case of the pure PVA, a co-peak in the same position as this peak can be present because of the incomplete hydrolyzation of the polymer at neutral pH, or reflecting water adsorption [40].

## Data Availability

The datasets used and/or analyzed during the current study are available from the corresponding author upon reasonable request.

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
