# Peer review of "The Effect of the PVA/Chitosan/Citric Acid Ratio on the Hydrophilicity of Electrospun Nanofiber Meshes"

_polymers, 2021, doi:10.3390/polym13203557_

Round 1

Reviewer 1 Report

The revised version of the manuscript addressed all my previous suggestions. After an in-depth English and grammar review, I believe that the manuscript will be ready for publication. I only suggest reviewing the conclusion section that might include some statements about the biocompatibility studies.

Author Response

Reviewer 1.

Comments and Suggestions for Authors

The revised version of the manuscript addressed all my previous suggestions. After an in-depth English and grammar review, I believe that the manuscript will be ready for publication. I only suggest reviewing the conclusion section that might include some statements about the biocompatibility studies.

Answers for Reviewer 1.

Dear Reviewer,

First of all we would like to thank for the review in-depth and for all comments and suggestions for improving our work.

Regarding your comments.

  1. English and grammar review will be asked from MDPI.

  1. The conclusion was improved according the the reviewers suggestion to the following:

“In our work we proved that beside the degree of cross-linkage the PVA/Chitosan/citric acid ratio has also an affect on the hydrophilicity and even on the cell viability and so on the biocompatibility of the fiber net. We optimized successfully the chemical composition and also the cross-linkage of the fibre net manifesting with appropriate hydrophilicity (H2 sample with contact angle: 42.5+4.29◦), as according to our results the original structure of heat treated scaffolds possibly will not changed significantly in aqueous media during the working and curing time. As the mesh (H2 containing 6.5 g/100g PVA, 1.3 g/100g chitosan and 2.9 g/100g citric acid) maintained all the necessary physical and chemical properties (for example: geometry, pore size, pore interconnectivity) of nanofibres and proved to be biocompatible with DPSC cells it can be potentially used for dental applications.”

I hope the modifications with the related answers are acceptable for You.

Yours faithfully,

Zsuzsanna Czibulya, PhD.

Department of Biomaterials and Prosthetic Dentistry Faculty of Dentistry, University of Debrecen,Egyetem tér 1., 4032 Debrecen, Hungary,Phone number: +36-52-411717/56970, Fax: +36-52-255208, email address: czibulya.zsuzsanna@dental.unideb.hu

Reviewer 2 Report

The manuscript is well written. It deals with evaluation of composite nanofiber mats - scaffolds for the medical application, specifically dental one. The authors presents PVA meshes with addition of chitosan and citric acid to improve stability of the composite electrospun mats without losing the hydrophilicity (some level) and thus preserving a good biocompatibility.

The research was carried out and presented in a systematic and comprehensive manner, with appropriate analysis and discussion of the results. However, the authors did not avoid a few errors and some parts could be improved, namely:

  • In lines 29, 191, 497, 505 there are typos.
  • In “2.1. materials” section (lines 114-116) deacetylation degree and molecular weight of chitosan are citated, which cannot be accepted. Chitosan, as natural material, differs from batch to batch, which, somehow, is one of the biggest disadvantage of this polymer. Considering the fact that biological properties (such as biocompatibility, antibacterial properties) are strongly influenced by chitosan DD, authors should conduct at least potentiometric measurement to determine this parameter. The measurement is easy and quick and do not request any of advance equipment or chemicals. Or in the future work consider to use chitosan recommended for biological/medical application which usually is well characterized by producer.
  • In line 158 is written: “as to our experiments chitosan containing samples melt point is 102.3 ◦C”. This part of sentence suggest that chitosan is responsible for melting, but chitosan do not have a melting point. For melting some other component of fibers has to be responsible, so the sentence is misleading.
  • Line 174: IR resolution should be rather written in cm-1,
    and deconvolution of spectra is better to perform with software dedicated to spectroscopy, as it takes into account specificity of spectroscopic measurements
  • 2.7 section (line 190): size of drop (mL) and the number of measurement repetitions are missing
  • Table 2 (line 250): Do two decimal places have a physical meaning? Shouldn't the numbers be rounded to one?
  • Figure 3a and 3b: Please enlarge the fibers images (but not diminishing the fiber size distribution charts)
  • Table 5 (line 337): Contact angle values should be rounded at least to the first decimal place due to the size of the measurement error
  • In line 347 is written: “To confirm the development of cross-linkage, therefore higher hydrophobicity, related to higher contact angle values Infrared spectra were measured.” – This sentence is quite arguable because contact angle values are also strongly governed by the roughness (here there should be described by Cassie-Baxter model), of course crosslinking is needed for stability/insolubility of samples but if one sample, for example H2 would be produced with various crosslinking degree by different thermal treatment it can still be the same CA and various IR spectra. I would rather agree just with confirming crosslinking by IR and separately in discussion mention about indirect correlation of crosslinking with hydrophobicity and higher contact angle.

Author Response

For Reviewer 2.

Dear Reviewer,

First of all we would like to thank for the review in-depth and for all comments and suggestions for improving our work. Answers will be given point by point to each suggestions.

  • In lines 29, 191, 497, 505 there are typos.

Authors thank you for the remarks, all typos were improved.

  • In “2.1. materials” section (lines 114-116) deacetylation degree and molecular weight of chitosan are citated, which cannot be accepted. Chitosan, as natural material, differs from batch to batch, which, somehow, is one of the biggest disadvantage of this polymer. Considering the fact that biological properties (such as biocompatibility, antibacterial properties) are strongly influenced by chitosan DD, authors should conduct at least potentiometric measurement to determine this parameter. The measurement is easy and quick and do not request any of advance equipment or chemicals. Or in the future work consider to use chitosan recommended for biological/medical application which usually is well characterized by producer.

Thank you for your suggestions, we realized the problem, and plan to do our further investigations by a more well characterized type of Chitosan, but recently we have measured the asked parameters, and the results with the related methods were desribe in materials and methods part.

“The molecular weight of chitosan was determined by size exclusion chromatography against polyethylene oxide molecular weight standards on a Phenomenex Polysep-P-Linear (300×7.8 mm) column using 0.1% aqueous trifluoroacecic acid as eluent. The system consisted of a Merck-Hitachi LaChrom D-7000 liquid chromatograph equipped with variable wavelength UV and differential refractive index detectors. The eluent flow rate was set to 0.7 ml/min and the column compartment was thermostated to 40 °C. Chitosan was found to have a peak molecular weight of 750-1000 kDa and broad polydispersity.

The degree of deacetylation was calculated from the 1H-NMR spectrum of 10 mg chitosan dissolved in 1 ml D2O containing 0.08 M DCl, by using the formula of Hirai et al. (Polymer Bulletin 26, 87-94, 1991). The chitosan sample was found to have a degree of deacetylation of 82%.”

In line 158 is written: “as to our experiments chitosan containing samples melt point is 102.3 ◦C”. This part of sentence suggest that chitosan is responsible for melting, but chitosan do not have a melting point. For melting some other component of fibers has to be responsible, so the sentence is misleading.

The reviewer has right, of course fibers can have a well-defined melting point. Authors made preliminary investigations on the used systems, and according to this they chose the applicated heat treatment procedure. Therefore, this sentence was modified.

  • Line 174: IR resolution should be rather written in cm-1,
    and deconvolution of spectra is better to perform with software dedicated to spectroscopy, as it takes into account specificity of spectroscopic measurements

The resolution was given in cm-1. I was 0.482cm-1.

We also, thank you for your suggestion, about deconvolution. The curves were deconvoluted by Origin Pro program, peak finder function, which was developed special for spectral analyses, it was used in our work, because this was also used in the first author’s previous works as the below mentioned one.

Luminescence, Sep-Oct 2013;28(5):726-33. doi: 10.1002/bio.2423. Epub 2012 Sep 17, Miklós Poór, Sándor Kunsági-Máté, Zsuzsanna Czibulya, Yin Li, Beáta Peles-Lemli, József Petrik, Sanda Vladimir-Knežević, Tamás Kőszegi, Fluorescence spectroscopic investigation of competitive interactions between ochratoxin A and 13 drug molecules for binding to human serum albumin

  • 2.7 section (line 190): size of drop (mL) and the number of measurement repetitions are missing

The asked parameters were inserted in adequate place. The size of drops was 0.004 mL and the number of measurement repetitions was 5.

  • Table 2 (line 250): Do two decimal places have a physical meaning? Shouldn't the numbers be rounded to one?

The reviewer has right, values in Table 2. were rounded to one.

  • Figure 3a and 3b: Please enlarge the fibers images (but not diminishing the fiber size distribution charts)

Figure 3a and 3b was modified as you can see in the attachment (in the doc file, by the related part).

  • Table 5 (line 337): Contact angle values should be rounded at least to the first decimal place due to the size of the measurement error

The table was corrected, contact angle values were rounded to the first decimal place.

  • In line 347 is written: “To confirm the development of cross-linkage, therefore higher hydrophobicity, related to higher contact angle values Infrared spectra were measured.” – This sentence is quite arguable because contact angle values are also strongly governed by the roughness (here there should be described by Cassie-Baxter model), of course crosslinking is needed for stability/insolubility of samples but if one sample, for example H2 would be produced with various crosslinking degree by different thermal treatment it can still be the same CA and various IR spectra. I would rather agree just with confirming crosslinking by IR and separately in discussion mention about indirect correlation of crosslinking with hydrophobicity and higher contact angle.

Row 347 was corrected. “To confirm the development of cross-linkage infrared spectra were measured.”

The discussion part was completed by the following sentences: “Cross-linking process can also be monitored by infrared spectroscopy, as there is an indirect correlation of crosslinking with hydrophobicity and higher contact angle. In case of fibers under some roughness conditions air bubbles can be trapped, and the contact angles increase, only apparent contact angles can be given. This property can be described by Cassie-Baxter state.”

I hope the modifications with the related answers are acceptable for You.

Yours faithfully,

Zsuzsanna Czibulya, PhD.

Department of Biomaterials and Prosthetic Dentistry Faculty of Dentistry, University of Debrecen,Egyetem tér 1., 4032 Debrecen, Hungary,Phone number: +36-52-411717/56970, Fax: +36-52-255208, email address: czibulya.zsuzsanna@dental.unideb.hu

This manuscript is a resubmission of an earlier submission. The following is a list of the peer review reports and author responses from that submission.

Round 1

Reviewer 1 Report

Peer review report on " The effect of the PVA/Chitosan/Citric acid ratio on the hydrophilicity of electrospun nanofiber meshes."

  1. Recommendation

 Major corrections

 Comments to the author:

Manuscript ID: polymers-1332429

Title: The effect of the PVA/Chitosan/Citric acid ratio on the hydrophilicity of electrospun nanofiber meshes

Overview and general recommendation:

The use of molecular scaffolds made of biopolymers has been established as an appropriate treatment method to improve multiple diseases and wounds. Among the most used biopolymers are PVA and chitosan through the electrospinning technique, which allows the precise and reproducible design of porous and flexible structures and is non-toxic and biocompatible. In this work, electrospun fibers of PVA-chitosan and citric acid were obtained in variable mass proportions. In addition, in the manuscript, applying a heat treatment after electrospinning to cross-link the polymers was analyzed.

In general, the proposal is exciting. However, there are many aspects to improve so that the quality of the manuscript increases.

There are errors related to the coherence of ideas and writing. Starting from the quality of the writing, the manuscript requires an exhaustive revision of the grammar, the use of punctuation marks, the correction of typing errors, and the excessive use of personal pronouns that reduce the elegance of the reading. Additionally, it is required to review the citation style used that does not correspond to that of the magazine.

The summary and introduction have objectives that cannot be supported by the results shown. Specifically, it is asserted that formulations were optimized for dental applications, but the requirements that a biomaterial must meet in these areas are never clarified. The use of fibers to release drugs, for the regeneration of pulp/teeth, and to cure dental caries is being considered. The spectrum is vast, and it is not clear how this formulation can achieve that just by increasing the hydrophobicity of the scaffolds. Moreover, the conclusions seriously affirm that the membranes maintained all the properties necessary for these applications, such as mechanical resistance, interconnectivity, pore size, and geometry, when only the fibers' diameters and morphology.

Also, it is necessary to clarify the suitability of the membranes for dental applications. Although cell viability did not decrease much (although the percentages are not determined) with the introduction of chitosan, it is stated that the morphology of DPSC cells and that perhaps that is the reason for lower adhesiveness, which already implies structural changes in cells and possibly in their intracellular content. Furthermore, the authors state that there are metabolic changes and do not explain how they reach this conclusion. Still, it could also affect the use of these membranes in biomedical applications such as those proposed.

The effect of the solution's viscosity (and therefore chitosan) in the electrospinning process needs in-depth analysis.

There are many other aspects highlighted throughout the article that must be clarified or corrected point by point, sent through the attached pdf.

For this article to be published, all comments and corrections must be applied or discussed.

Reviewer 2 Report

The work deals with the fabrication and characterization of composite PVA/chitosan/citric acid electrospun nanofibers with potential applications for biomedical and pharmaceutical products. Despite the high number of characterization techniques carried out by the Authors, this research lacks novelty and originality since several similar and more complete articles have already been published in recent years. Additionally, besides the conclusion drawn out by the Authors may appear interesting, some of them are not totally confirmed by the experimental results and they could even have a different explanation.

In my opinion, the results are publishable in Polymers journal but the work needs to be completely revised, more experiments must be performed, and the results should be discussed more in detail. Hence, I would suggest the Authors to resubmit the work after all these concerns have been considered.

Below, a point-by-point list of the aspects that should be addressed by the Authors is reported:

  1. In the Abstract the role of citric acid should be clearly indicated.
  2. Lines 39-50

The Authors are way too vague in their description of electrospinning and biomaterials. Specifically, they should better report the requirements for scaffolding materials, also making some relevant examples, and the drawbacks related to the use of biomaterials in combination with electrospinning.

  1. Lines 53-55

PVA water solubility at room temperature is actually quite low, especially for high molecular weight polymers. Additionally, humble thermal treatments can be applied to stabilize the fibers in aqueous environments. In this sense, the Authors should clarify why pristine PVA electrospun fibers are not suited for the targeted application.

  1. Lines 66-67

I am not completely convinced about the requirement of a crosslinking treatment to stabilize PVA/chitosan fibers. Being chitosan quite hydrophobic, mixing it with PVA should be enough to stabilize the mat. Additional tests using pristine PVA, pristine chitosan, and pristine PVA/chitosan fibers are required to prove the significance of the proposed approach.

  1. Section 2.2.3

Did the Author add any initiator? It should be of a great importance in order to induce the crosslinking reaction via citric acid.

  1. Lines 138-139

Such melting temperatures are low compared to what is expected. Can the Authors add the DSC curves?

  1. Lines 140-142

Did the Authors observe a change in the sample colour upon the heating treatment?

  1. Line 163

The Authors should clarify if they measured the initial or the final contact angle.

  1. Section 3.1

How did the Authors calculate the mixture viscosity? Which formula has been used? Do the flow curves present the expected behaviour for polymeric systems? Please provide more details.

  1. Figure 3

Please rethink the Figure. It is difficult to observe the fibers due to the dimension of the insets.

  1. Lines 220-221

This may simply related to the higher viscosity of the electrospun mixture. The Authors may try to modify the processing parameters in order to get rid of such defects.

  1. Lines 254-257

Why did AFM measurements allow to monitor the changes in the fiber diameter whereas the same was not possible via SEM? If SEM analysis is more precise than AFM images, the results are not expected to be reliable.

  1. Lines 266-268

Can the Authors provide references where the roughness measured via AFM is related to the fiber density in electrospun mats?

  1. Section 3.4

The water contact angle should be measured also before the heat treatment for better comparison, it’s very unlikely that PVA/chitosan mats dissolve in water so fastly.

  1. Lines 320-327

Despite the error in the swelling experiments for electrospun mats is always high, the data showed by the authors seems to present a well-defined trend as a function of citric acid content. Can the Authors better comment on such a finding?